# FEATURE-WISE BIAS AMPLIFICATION

**Klas Leino, Matt Fredrikson, Emily Black, Shayak Sen, & Anupam Datta**
Carnegie Mellon University

## ABSTRACT

We study the phenomenon of *bias amplification* in classifiers, wherein a machine learning model learns to predict classes with a greater disparity than the underlying ground truth. We demonstrate that bias amplification can arise via an inductive bias in gradient descent methods that results in the overestimation of the importance of moderately-predictive "weak" features if insufficient training data is available. This overestimation gives rise to *feature-wise bias amplification* – a previously unreported form of bias that can be traced back to the features of a trained model. Through analysis and experiments, we show that while some bias cannot be mitigated without sacrificing accuracy, feature-wise bias amplification can be mitigated through targeted feature selection. We present two new feature selection algorithms for mitigating bias amplification in linear models, and show how they can be adapted to convolutional neural networks efficiently. Our experiments on synthetic and real data demonstrate that these algorithms consistently lead to reduced bias without harming accuracy, in some cases eliminating predictive bias altogether while providing modest gains in accuracy.

## 1 INTRODUCTION

*Bias amplification* occurs when the distribution over prediction outputs is skewed in comparison to the prior distribution of the prediction target. Aside from being problematic for accuracy, this phenomenon is also potentially concerning as it relates to the *fairness* of a model's predictions (Zhao et al., 2017; Burns et al., 2018; Bolukbasi et al., 2016; Stock & Cissé, 2017) as models that learn to overpredict negative outcomes for certain groups may exacerbate stereotypes, prejudices, and disadvantages already reflected in the data (Hart, 2017).

Several factors can cause bias amplification in practice. The *class imbalance problem* is a well-studied scenario where some classes in the data are significantly less likely than others (Wallace et al., 2011a). Classifiers trained to minimize empricial risk are not penalized for ignoring minority classes. However, as we show through analysis and experiments, bias amplification can arise in cases where the class prior is not severely skewed, or even when it is unbiased. Thus, techniques for dealing with class imbalance alone cannot explain or address all cases of bias amplification.

We examine bias amplification in the context of binary classifiers, and show that it can be decomposed into a component that is intrinsic to the model, and one that arises from the inductive bias of gradient descent on certain feature configurations. The intrinsic case manifests when the class prior distribution is more informative for prediction than the features, causing the the model to predict the class mode. This type of bias is unavoidable, as we show that any mitigation of it will lead to less accurate predictions (Section 3.1).

Interestingly, linear classifiers trained with gradient descent tend to overestimate the importance of moderately-predictive, or "weak," features if insufficient training data is available (Section 3.2). This overestimation gives rise to *feature-wise bias amplification* – a previously unreported form of bias (see Section 2 for comparison to related work) that can be traced back to the features of a trained model. It occurs when there are more features that positively correlate with one class than the other. If these features are given undue importance in the model, then their combined influence will lead to bias amplification in favor of the corresponding class.

A concrete example of this would be predicting gender from face images. Suppose that there are several features that occur slightly more often with women (i.e. are weakly correlated with women) – long hair, eyeliner, purses, smiling, necklaces – but only a few that are more correlated with

men – necktie, mustache. Then, our analysis suggests a model would exhibit feature-wise bias amplification towards female prediction, because there are more weak features oriented towards that class. In fact, we experimentally demonstrate that feature-wise bias amplification can happen even when the class prior is unbiased; e.g., even if the aforementioned hypothetical classification problem contained an equal number of male and female training instances.

Our analysis sheds new light on real instances of the problem, and paves the way for practical mitigations of it. The existence of such moderately-predictive weak features is not uncommon in models trained on real data. Viewing deep networks as the composition of a feature extractor and a linear classifier, we explain some instances of bias amplification in deep networks (Table 1, Section 5).

Finally, this understanding of feature-wise bias amplification motivates a solution based on feature selection. We develop two new feature-selection algorithms that are designed to mitigate bias amplification (Section 4). We demonstrate their effectiveness on both linear classifiers and deep neural networks (Section 5). For example, for a VGG16 network trained on CelebA (Liu et al., 2015) to predict the "attractive" label, our approach removed 95% of the bias in predictions. We observe that in addition to mitigating bias amplification, these feature selection methods reduce generalization error relative to an $\ell_1$ regularization baseline for both linear models and deep networks (Table 1).

## 2  RELATED WORK

While the term bias is used in a number of different contexts in machine learning, we use *bias amplification* in the sense of Zhao et al. (2017), where the distribution over prediction outputs is skewed in comparison to the prior distribution of the prediction target. For example, Zhao et al. (2017) and Burns et al. (2018) use the imSitu vSRL dataset for the MS-COCO task, i.e. to classify agents and actions in pictures. In the dataset, women are twice as likely to be the agent when the action is cooking, but the model was five times as likely to predict women to be the agent cooking.

In a related example, Stock & Cissé (2017) identify bias in models trained on the ImageNet dataset. Despite there being near-parity of white and black people in pictures in the basketball class, 78% of the images that the model classified as *basketball* had black people in them and only 44% had white people in them. Additionally, 90% of the misclassified *basketball* pictures had white people in them, whereas only 20% had black people in them. Note that this type of bias over classes is distinct from the learning bias in machine learning (Geman et al., 1992) which has received renewed interest in the context of SGD and under-determined models (Gunasekar et al., 2018; Soudry et al., 2017).

Naturally, bias has been studied in the context of fairness, and in particular in cases where pre-existing societal biases are propagated by a model's predictions. One recent example of this is Amazons gender-biased recruiting tool, which learned not to prefer candidates from all-female colleges based on past hiring data(Dastin, 2018). Removing such biases from models' behavior altogether using fairness criteria (Bilal Zafar et al., 2015; Binns, 2018; Zliobaite, 2015) requires deviating from patterns in the data, and may conflict with accuracy. However, bias can also come as a product of the model itself: a model could ostensibly take a relatively unbiased dataset and erroneously make more biased predictions due to some limitations of learning. Feature-wise bias falls into this class, as it is created by the mechanics of the learning rule itself rather than stemming from the data. This means that removing feature-wise bias would make the model more faithful to the data it was given, generally increasing accuracy. Consequently, this paper does not weigh in on which fairness guidelines to use in navigating the bias-accuracy tradeoff, since it does not apply to the type of bias we present.

Bias amplification is often thought to be result of class imbalance in the training data, which is well-studied in the learning community (see He & Garcia (2009) and Buda et al. (2017) for comprehensive surveys). There are a myriad of empirical investigations of the effects of class imbalance in machine learning and different ways of mitigating these effects (Maloof, 2003; Chawla, 2005; Mazurowski et al., 2008; Oommen et al., 2011; Wallace et al., 2011b).

It has been shown that neural networks are affected by class imbalance as well (Murphey et al., 2004). Buda et al. (2017) point out that the detrimental effect of class imbalance on neural networks increases with scale. They advocate for an oversampling technique mixed with thresholding to improve accuracy based on empirical tests. An interesting and less common technique from Havaei et al. (2015) relies on a drastic change to neural network training procedure in order to better detect

brain tumors: they first train the net on an even distribution, and then on a representative sample, but only on the output layer in the second half of training.

In contrast to prior work, we demonstrate that bias amplification can occur without existing imbalances in the training set. Therefore, we identify a new source of bias that can be traced to particular features in the model. Since we remove bias feature-wise, our approach can also be viewed as method for feature selection. While feature selection is a well-studied problem, to the authors' knowledge, no one has looked at removing features to mitigate *bias*. Generally, feature selection has been applied for improving model accuracy, or gaining insight into the data (Chandrashekar & Sahin, 2014). For example, Kim et al. (2015) use feature selection for interpretability during data exploration. They select features that have high variance across clusters created based on human-interpretable, logical rules. Differing from prior work, we focus on bias by identifying features that are likely to increase bias, but can be removed while maintaining accuracy.

Naive Bayes classification models comprise a similarly well-studied topic. Rennie et al. (2003) point out common downfalls of Naive Bayes classifiers on datasets that do not meet Naive Bayes criteria: bias from class imbalance, and the problem of over-predicting classes with correlated features. Our work shows that similar effects can occur even on data that *does* match Naive Bayes assumptions. Zhang (2004) shows that the naive Bayes classifier is optimal so long as the dependencies between features over the whole network cancel each other out. Our work can mitigate bias in scenarios where these conditions do not hold.

## 3 BIAS AMPLIFICATION IN BINARY CLASSIFIERS

In this section, we define bias amplification for binary classifiers, and show that in some cases it may be unavoidable. Namely, a Bayes-optimal classifier trained on poorly-separated data can end up predicting one label nearly always, even if the prior label bias is minimal. While our analysis makes strong generative assumptions, we show that its results hold qualitatively on real data that resemble these assumptions. We begin by formalizing the setting.

We consider the standard binary classification problem of predicting a label $y \in \{0, 1\}$ given features $\mathbf{x} = (x_1, \ldots, x_d) \in \mathcal{X}$. We assume that data are generated from some unknown distribution $\mathcal{D}$, and that the prior probability of $y = 1$ is $p^*$. Without loss of generality, we assume that $p^* \geq 1/2$. The learning algorithm recieves a training set $S$ drawn i.i.d. from $\mathcal{D}^n$ and outputs a predictor $h_S : \mathcal{X} \to \{0, 1\}$ with the goal of minimizing 0-1 loss on unknown future i.i.d. samples from $\mathcal{D}$.

**Definition 1 (Bias amplification, systematic bias)** *Let $h_S$ be a binary classifier trained on $S \sim \mathcal{D}^n$. The bias amplification of $h_S$ on $\mathcal{D}$, written $B_{\mathcal{D}}(h_S)$, is given by Equation 1.*

$$B_{\mathcal{D}}(h_S) = \mathop{\mathbb{E}}_{(\mathbf{x},y)\sim\mathcal{D}} [h_S(\mathbf{x}) - y] \tag{1}$$

*We say that a learning rule exhibits* systematic bias *whenever it exhibits non-zero bias amplification on average over training samples, i.e. it satisfies Equation 2.*

$$\mathop{\mathbb{E}}_{S\sim\mathcal{D}^n} [B_{\mathcal{D}}(h_S)] \neq 0 \tag{2}$$

Definition 1 formalizes bias amplification and systematic bias in this setting. Intuitively, bias amplification corresponds to be the probability that $h_S$ predicts class 1 on instances from class 0 in excess of the prior $p^*$. Systematic bias lifts the definition to learners, characterizing rules that are expected to amplify bias on training sets drawn from $\mathcal{D}$.

### 3.1 SYSTEMATIC BIAS IN BAYES-OPTIMAL PREDICTORS

Definition 1 makes it clear that systematic bias is a property of the learning rule producing $h_S$ and the distribution, so any technique that aims to address it will need to change one or both. However, if the learner always produces Bayes-optimal predictors for $\mathcal{D}$, then any such change will result in suboptimal classifiers, making bias amplification *unavoidable*. In this section we characterize the systematic bias of a family of linear Bayes-optimal predictors.

Consider a special case of binary classification in which $\mathbf{x}$ are drawn from a multivariate Gaussian distribution with class means $\boldsymbol{\mu}_0^*, \boldsymbol{\mu}_1^* \in \mathbb{R}^d$ and diagonal covariance matrix $\boldsymbol{\Sigma}^*$, and $y$ is a Bernoulli

| dataset | $D$ | $p^*$ | $B_\mathcal{D}(h_S)$ | % acc |
|---|---|---|---|---|
| banknote | 1.87 | 0.56 | 0.04 | 84.1 |
| breast cancer wisc | 1.81 | 0.63 | 0.02 | 94.2 |
| drug consumption | 0.86 | 0.78 | 0.12 | 75.6 |
| pima diabetes | 1.15 | 0.66 | 0.08 | 79.9 |

(a)

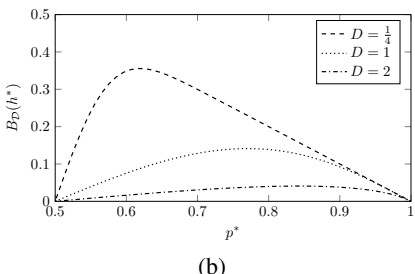

(b)

Figure 1: (a) Bias amplification on real datasets classified using Gaussian Naive Bayes; (b) bias amplification of Bayes-optimal classifier in terms of the Mahalanobis distance $D$ between class means and prior class probability $p^*$.

random variable with parameter $p^*$. Then $\mathcal{D}$ is given by Equation 3.

$$\mathcal{D} \triangleq \Pr[\mathbf{x}|y] = \mathcal{N}(\mathbf{x}|\boldsymbol{\mu}_y^*, \boldsymbol{\Sigma}^*), y \sim \text{Bernoulli}(p^*) \tag{3}$$

Because the features in $\mathbf{x}$ are independent given the class label, the Bayes-optimal learning rule for this data is Gaussian Naive Bayes, which is expressible as a linear classifier (Murphy, 2012).

Making the ideal assumption that we are always able to learn the Bayes-optimal classifier $h^*$ for parameters $\boldsymbol{\mu}_y^*, \boldsymbol{\Sigma}^*, p^*$, we proceed with the question: does $h^*$ have systematic bias? Our assumption of $h_S = h^*$ reduces this question to whether $B_\mathcal{D}(h^*)$ is zero. Proposition 1 shows that $B_\mathcal{D}(h^*)$ is strictly a function of the class prior $p^*$ and the Mahalanobis distance $D$ of the class means $\boldsymbol{\mu}_y^*$. Corollary 1 shows that when the prior is unbiased, the model's predictions remain unbiased.

**Proposition 1** *Let $\mathbf{x}$ be distributed according to Equation 3, $y$ be Bernoulli with parameter $p^*$, $D$ be the Mahalanobis distance between the class means $\boldsymbol{\mu}_0^*, \boldsymbol{\mu}_1^*$, and $\beta = -D^{-1} \log(p^*/(1-p^*))$. Then the bias amplification of the Bayes-optimal classifier $h^*$ is:*

$$B_\mathcal{D}(h^*) = 1 - p^* - (1-p^*)\Phi\left(\beta + \tfrac{D}{2}\right) - p^*\Phi\left(\beta - \tfrac{D}{2}\right)$$

**Corollary 1** *When $\mathbf{x}$ is distributed according to Equation 3 and $p^* = 1/2$, $B_\mathcal{D}(h^*) = 0$.*

The proofs of both claims are given in the appendix. Corollary 1 is due to the fact that when $p^* = 1/2$, $\beta = 0$. Because of the symmetry $\Phi(-x) = 1 - \Phi(x)$, the $\Phi$ terms cancel out giving $\Pr[h^*(\mathbf{x}) = 1] = 1/2$, and thus the bias amplification $B_\mathcal{D}(h^*) = 0$.

Figure 1a shows the effect on real data available on the UCI repository classified using Gaussian Naive Bayes (GNB). These datasets were chosen because their distributions roughly correspond to the naive Bayes assumption of conditional feature independence, and GNB outperformed logistic regression. In each case, bias amplification occurs in approximate correspondence with Proposition 1, tracking the empirical class prior and class distance to Figure 1b.

Figure 1b shows $B_\mathcal{D}(h^*)$ as a function of $p^*$ for several values of $D$. As the means grow closer together, there is less information available to make reliable predictions, and the label prior is used as the more informative signal. Note that $B_\mathcal{D}(h^*)$ is bounded by 1/2, and the critical point corresponds to bias "saturation" where the model always predicts class 1. From this it becomes clear that the extent to which overprediction occurs grows rather quickly when the means are moderately close. For example when $p^* = 3/4$ and the class means are separated by distance $1/2$, the classifier will predict $Y = 1$ with probability close to 1.

*Summary*: Bias amplification may be *unavoidable* when the learning rule is a good fit for the data, but the features are less effective at distinguishing between classes than the prior. Our results show that in the particular case of conditionally-independent Gaussian data, the Bayes-optimal predictor suffers from bias as the Mahalanobis distance between class means decreases, leading to a noticeable increase even when the prior is only somewhat biased. The effect is strong enough to manifest in real settings where generative assumptions do not hold, but GNB outperforms other linear classifiers.

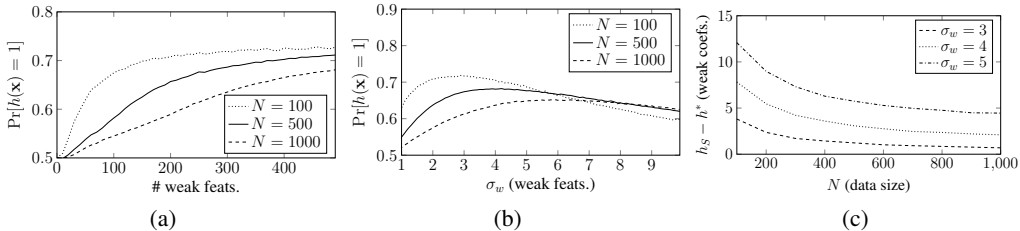

Figure 2: (a), (b): Expected bias as a function of (a) number of weak features and (b) variance of the weak features, shown for models trained on $N = 100, 500, 1000$ instances. $\sigma_w$ in (a) is fixed at 10, and in (b) the number of features is fixed at 256. (c): Extent of overestimation of weak-feature coefficients in logistic classifiers trained with stochastic gradient descent, in terms of the amount of training data. The vertical axis is the difference in magnitude between the trained coefficient ($h_S$) and that of the Bayes-optimal predictor ($h^*$). In (a)-(c), data is generated according to Equation 4 with $\sigma_s = 1$, and results are averaged over 100 training runs.

## 3.2 FEATURE ASYMMETRY AND GRADIENT DESCENT

When the learning rule does not produce a Bayes-optimal predictor, it may be the case that excess bias can safely be removed without harming accuracy. To support this claim, we turn our attention to logistic regression classifiers trained using stochastic gradient descent. Logistic regression predictors for data generated according to Equation 3 converge in the limit to the same Bayes-optimal predictors studied in Proposition 1 and Corollary 1 (Murphy, 2012).

Logistic regression models make fewer assumptions about the data and are therefore more widely-applicable, but as we demonstrate in this section, this flexibility comes at the expense of an inductive bias that can lead to systematic bias in predictions. To show this, we continue under our assumption that $\mathbf{x}$ and $y$ are generated according to Equation 3, and consider the case where $p^* = 1/2$. According to Corollary 1, any systematic bias that emerges must come from differences between the trained classifier $h_S$ and the Bayes-optimal $h^*$.

### 3.2.1 FEATURE ASYMMETRY

To define what is meant by "feature asymmetry", consider the orientation of each feature $x_j$ as given by the sign of $\mu_{1j} - \mu_{0j}$. The sign of each coefficient in $h^*$ will correspond to its feature orientation, so we can think of each feature as being "towards" either class 0 or class 1. Likewise, we can view the combined features as being *asymmetric towards* $y$ when there are more features oriented towards $y$ than towards $1 - y$.

As shown in Table 1 (Section 5), high-dimensional data with biased class priors often exhibit feature asymmetry towards the majority class. This does not necessarily lead to excessive bias, and the analysis from the previous section indicates that if $p^* = 1/2$ then it may be possible to learn a predictor with no bias. However, if the learning rule overestimates the importance of some of the features oriented towards the majority class, then variance in those features present in minority instances will cause mispredictions that lead to excess bias beyond what is characterized in Proposition 1.

This problem is pronounced when many of the majority-oriented features are weak predictors, which in this setting means that the magnitude of their corresponding coefficients in $h^*$ are small relative to the other features (for example, features with high variance or similar means between classes). The weak features have small coefficients in $h^*$, but if the learner systematically overestimates the corresponding coefficients in $h_S$, the resulting classifier will be "out of balance" with the distribution generating the data.

Figure 2 explores this phenomenon through synthetic Gaussian data exemplifying this feature asymmetry, in which the strongly-predictive features have low variance $\sigma_s = 1$, and the weakly-predictive features have relatively higher variance $\sigma_w > 1$. Specifically, the data used here follows Equation 3 with the parameters shown in Equation 4.

$$p^* = 1/2, \boldsymbol{\mu}_0^* = (0, 1, 0, \ldots, 0), \boldsymbol{\mu}_1^* = (1, 0, 1, \ldots, 1), \boldsymbol{\Sigma}^* = \mathrm{diag}(\sigma_s, \sigma_s, \sigma_w, \ldots, \sigma_w) \quad (4)$$

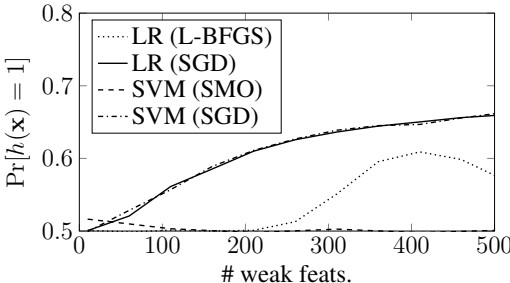

Figure 3: Bias from linear classifiers on data generated according to Equation 4 with $\sigma_s = 1$ (i.e., generated in the same manner as the experiments in Figure 2), averaged over 100 training runs. The SVM trained using SMO used penalty $C = 1.0$ and the linear kernel. Regardless of the loss used, the bias of classifiers trained using SGD is uniform and consistent, increasing with feature asymmetry. Comparable classifiers trained using other methods are not consistent in this way. While LR trained with L-BFGS does exhibit bias, it is not as strong, and does not appear in as many data configurations, as LR trained with SGD. While linear SVM with penalty trained with SMO results in little bias, SVM trained with SGD shows the same bias as LR. Not shown are results for classifiers trained with SGD using modified Huber, squared hinge, and perceptron losses, all of which closely match the two curves shown here for SGD classifiers.

Figure 2c suggests that overestimation of weak features is precisely the form of inductive bias exhibited by gradient descent when learning logistic classifiers. As $h_S$ converges to the Bayes-optimal configuration, the magnitude of weak-feature coefficients gradually decreases to the appropriate quantity. As the variance increases, the extent of the overapproximation grows accordingly.

While this effect may arise when methods other than SGD are used to estimate the coefficients, Figure 3 shows that it occurs consistently in models trained using SGD. In particular, we discovered the same bias amplification effect in the asymmetric feature setting when using SGD on multiple different loss functions. This suggests that this effect is brought about by SGD, and not the logistic loss in particular. Furthermore, our results in Table 1 (Section 5) reveal that feature-wise bias amplification can occur in deep models, i.e., the phenomenon is not limited to linear models.

### 3.2.2 PREDICTION BIAS FROM INDUCTIVE BIAS

While the classifier remains far from convergence, the cumulative effect of feature overapproximation with high-dimensional data leads to systematic bias. Figure 2a demonstrates that as the disparity in weak features towards class $y = 1$ increases, so does the expected bias towards that class. This bias cannot be explained by Proposition 1, because this data is distributed with $p^* = 1/2$. Rather, it is clear that the effect diminishes as the training size increases and $h_S$ converges towards $h^*$. This suggests gradient descent tends to "overuse" the weak features prior to convergence, leading to systematic bias that over-predicts the majority class in asymmetric regimes.

Figure 2b demonstrates that for a fixed disparity in weak features, the features must be sufficiently weak in order to cause bias. This suggests that a feature imbalance alone is not sufficient for causing systematic bias. Moreover, the weak features, rather than the strong features, are responsible for the bias. As the training size increases, the amount of variance required to cause bias increases. However, when the features have sufficiently high variance, the model will eventually decrease their contribution, relieving their impact on the bias and accuracy of the model.

*Summary*: When the data is distributed asymmetrically with respect to features' orientation towards a class, gradient descent may lead to systematic bias especially when many of the asymmetric features are weak predictors. This bias is a result of the learning rule, as it manifests in cases where a Bayes-optimal predictor would exhibit no bias, and therefore it may be possible to mitigate it without harming accuracy.

# 4 MITIGATING FEATURE-WISE BIAS AMPLIFICATION

While Theorem 1 suggests that some bias is unavoidable, the empirical analysis in the previous section shows that some systematic bias may not be. Our analysis also suggests an approach for removing such bias, namely by identifying and removing the weak features that are systematically overestimated by gradient descent. In this section, we describe two approaches for accomplishing this that are based on measuring the influence (Leino et al., 2018) of features on trained models. In Section 5, we show that these methods are effective at mitigating bias without harming accuracy on both logistic predictors and deep networks.

## 4.1 INFLUENCE-DIRECTED FEATURE REMOVAL

Given a model $h : \mathcal{X}_0 \to \mathbb{R}$ and feature, $x_j$, the *influence* $\chi_j$ of $x_j$ on $h$ is a quantitative measure of feature $j$'s contribution to the output of $h$. To extend this notion to internal layers of a deep network $h$, we consider the *slice abstraction* (Leino et al., 2018) comprised of a pair of functions $f : \mathcal{X}_0 \to \mathcal{X}$, and $g : \mathcal{X} \to \mathbb{R}$, such that $h = g \circ f$. We define $f$ to be the network up to the penultimate layer, and $g$ be the final layer. Intuitively, We can then think of the features as being precomputed by $f$, i.e., $\mathbf{x} = f(\mathbf{x}_0)$ for $\mathbf{x}_0 \in \mathcal{X}_0$, allowing us to treat the final layer as a linear model acting on features computed via a deep network. Note that the slice abstraction encompasses linear models as well, by defining $f$ to be the identity function.

A growing body of work on influence measures (Simonyan et al., 2013; Sundararajan et al., 2017; Leino et al., 2018) provides numerous choices for $\chi_j$, each with different tradeoffs. We use the *internal distributional influence* (Leino et al., 2018), as it incorporates the slice abstraction naturally. This measure is given by Equation 5 for a *distribution of interest* $P$, which characterizes the distribution of test instances.

$$\chi_j(g \circ f, P) = \int\limits_{\mathbf{x} \in \mathcal{X}_0} \left. \frac{\partial g}{\partial f(\mathbf{x})_j} \right|_{f(\mathbf{x})} P(\mathbf{x}) d\mathbf{x} \tag{5}$$

We now describe two techniques that use this measure to remove features causing bias.

**Feature parity.** Motivated by the fact that bias amplification may be caused by feature asymmetry, we can attempt to mitigate it by enforcing parity in features across the classes. To avoid removing features that are useful for correct predictions, we order the features by their influence on the model's output, and remove features from the majority class until parity is reached. If the model has a bias term, we adjust it by subtracting the product of each removed coefficient and the mean of its corresponding feature.

**Experts** Section 3.2 identifies "weak" features as a likely source of systematic bias. This is a somewhat artificial construct, as real data often does not exhibit a clear separation between strong and weak features. Qualitatively, the weak features are less predictive than the strong features, and the learner accounts for this by giving less influence to the weak features. Thus, we can think of imposing a strong/weak feature dichotomy by defining the weak features to be those such that $|\chi_j| < \chi^*$ for some threshold $\chi^*$. This reduces the feature selection problem to a search for an appropriate $\chi^*$ that mitigates bias to the greatest extent without harming accuracy.

We parameterize this search problem in terms of $\alpha, \beta$, where the $\alpha$ features with the most positive influence and $\beta$ features with the most negative influence are "strong", and the rest are considered weak. This amounts to selecting the *class-wise expert* (Leino et al., 2018) for the dominant class. Formally, let $F_\alpha$ be the set of $\alpha$ features with the $\alpha$ highest positive influences, and $F_\beta$ the set of $\beta$ features with the $\beta$ most negative influences. For slice $h = g \circ f$, let $g_\beta^\alpha$ be defined as model $g$ with its weights replaced by $w_\beta^\alpha$ as defined by Equation 6. Then we define *expert*, $g_{\beta*}^{\alpha^*}$, to be the classifier given by setting $\alpha^*$ and $\beta^*$ according to Equation 7. In other words, the $\alpha$ and $\beta$ that minimize bias while maintaining at least the original model's accuracy. Here $L_S$ represents the 0-1 loss on the training set, $S$.

$$\mathbf{w}_{\beta\,j}^\alpha = \begin{cases} \mathbf{w}_j & j \in F_\alpha \cup F_\beta \\ 0 & j \notin F_\alpha \cup F_\beta \end{cases} \tag{6}$$

$$\alpha^*, \beta^* = \arg\min_{\alpha,\beta} \left| B_\mathcal{D}(g_\beta^\alpha) \right| \text{ subject to } L_S(g_\beta^\alpha) \leq L_S(g) \tag{7}$$

| dataset | $p^*$ (%) | asymm. (%) | $B_\mathcal{D}(h_S)$ (%) | $B_\mathcal{D}(h_S)$ (%) (post-fix) | | | acc. (%) | acc. (%) (post-fix) | | |
|---|---|---|---|---|---|---|---|---|---|---|
| | | | | par | exp | $\ell_1$ | | par | exp | $\ell_1$ |
| CIFAR10 | 50.0 | 52.0 | 1.8 | 1.7 | **0.4** | n/a | 93.0 | 93.1 | **94.0** | n/a |
| CelebA | 50.4 | 50.2 | 7.7 | 7.7 | **0.2** | n/a | 79.6 | 79.6 | **79.9** | n/a |
| arcene | 56.0 | 57.7 | 2.7 | **0.6** | 1.2 | 1.7 | 68.9 | 69.0 | **74.2** | 69.4 |
| colon | 64.5 | 51.0 | 23.1 | 22.9 | **22.6** | 35.5 | 58.5 | 58.7 | 58.7 | **64.5** |
| glioma | 69.4 | 54.8 | 17.4 | 17.4 | **12.2** | 17.0 | 76.3 | 76.3 | **76.7** | 75.44 |
| micromass | 69.0 | 54.1 | 0.68 | **0.66** | 0.69 | 0.68 | **98.4** | **98.4** | **98.4** | **98.4** |
| pc/mac | 50.5 | 60.6 | 1.6 | 1.6 | **1.4** | 1.6 | **89.0** | **89.0** | 88.0 | **89.0** |
| prostate | 51.0 | 44.4 | 47.3 | 47.2 | **9.8** | 28.1 | 52.7 | 52.8 | **90.2** | 71.3 |
| smokers | 51.9 | 50.4 | 47.4 | 45.4 | **8.0** | 33.0 | 50.0 | 50.7 | **59.0** | 51.2 |
| synthetic | 50.0 | 99.9 | 24.1 | 17.2 | 23.6 | **5.7** | 74.9 | **77.9** | 74.8 | 71.4 |

Table 1: Bias measured on real datasets, and results of applying one of three mitigation strategies: feature parity (*par*), influence-directed experts (*exp*), and $\ell_1$ regularization. The columns give: $p^*$, percent class prior for the majority class ($y = 1$); *asymm*, the percentages of features oriented towards $y = 1$; $B_\mathcal{D}(h_S)$ the bias of the learned model on test data, which we measure before and after each fix (*post-fix*); *acc*, the test accuracy before and after each fix. The first two rows are experiments on deep networks, and the remainder are on 20 training runs of logistic regression with stochastic gradient descent. $\ell_1$ regularization was not applied to the deep network experiments due to the cost of hyperparameter tuning.

We note that this is always feasible by selecting all the features. Furthermore, this is a discrete optimization problem, which can be solved efficiently with a grid search over the possible $\alpha$ and $\beta$. In practice, even when there are many features, we can exhaustively search this space. When there are ties we can break them by preferring the model with the greatest accuracy.

## 5 EXPERIMENTS

In this section we present empirical evidence to support our claim that feature-wise bias amplification can safely be removed without harming the accuracy of the classifier. We show this on both logistic predictors and deep networks by measuring the bias on several benchmark datasets, and running the parity and expert mitigation approaches described in Section 4. As a baseline, we compare against $\ell_1$ regularization in the logistic classifier experiments.

The results are shown in Table 1. To summarize, on every dataset we consider, at least one of the methods in Section 4 proves effective at reducing the classifier's bias amplification. $\ell_1$ regularization removes bias less reliably, and never to the extent that our methods do. In all but two cases, the influence-directed experts show the best performance in terms of bias removal, and this method is able to reduce bias in all but one case. In terms of accuracy, our methods consistently improve classifier performance, and in some cases significantly. For example, on the *prostate* dataset, influence-directed experts removed 80% of the prediction bias while improving accuracy from 57.7% to 90.2%.

**Data.** We performed experiments over eight binary classification datasets from various domains (rows 3-11 in Table 1) and two image classification datasets (CIFAR10-binary, CelebA). Table 2 in the Appendix details the number of features and instances for each dataset. Our criteria for selecting logistic regression datasets were: high feature dimensionality, binary labels, and row-structured instances (i.e., not time series data). Among the logistic regression datasets, *arcene*, *colon*, *glioma*, *pc/mac*, *prostate*, *smokers* were obtained from the scikit-feature repository (Li et al., 2016), and *micromass* was obtained from the UCI repository (Dheeru & Karra Taniskidou, 2017). The synthetic dataset was generated in the manner described in Section 3.2, containing one strongly-predictive feature ($\sigma^2 = 1$) for each class, 1,000 weak features ($\sigma^2 = 3$), and $p^* = 1/2$.

For the deep network experiments, we created a binary classification problem from CI-FAR10 (Krizhevsky & Hinton, 2009) from the "bird" and "frog" classes. We selected these classes as they showed the greatest posterior disparity on VGG16 network trained on the original dataset. For CelebA, we trained a VGG16 network with one fully-connected layer of 4096 units to predict the *attractiveness* label given in the training data.

**Methodology.**  For the logistic regression experiments, we used scikit-learn's SGDClassifier estimator to train each model using the logistic loss function. Logistic regression measurements were obtained by averaging over 20 pseudorandom training runs on a randomly-selected stratified train/test split. Experiments involving experts selected $\alpha, \beta$ using grid search over the possible values that minimize bias subject to not harming accuracy as described in Section 4. Similarly, experiments involving $\ell_1$ regularization use a grid search to select the regularization paramter, optimizing for the same criteria used to select $\alpha, \beta$. Experiments on deep networks use the training/test split provided by the respective dataset authors. Models were trained until convergence using Keras 2 with the Theano backend.

**Logistic regression.**  Table 1 shows that on linear models, feature parity always improves or maintains the model in terms of both bias amplification and accuracy. Notably, in each case where feature parity removes bias, the accuracy is likewise improved, supporting our claim that bias resulting from asymmetric feature regimes is avoidable. In most cases, the benefit from applying feature parity is, however, rather small. *arcene* is the exception, which is likely due to the fact that it has large feature asymmetry in the original model, leaving ample opportunity for improvement by this approach.

The results suggest that influence-directed experts are the most effective mitigation technique, both in terms of bias removal and accuracy improvement. In most datasets, this approach reduced bias while improving accuracy, often substantially. Most notably on the *prostate* dataset, where the original model failed to achieve accuracy appreciably greater than chance and extreme bias. The mitigation achieves 90% accuracy while removing 80% of the bias, improving the model significantly. Similarly, for *arcene* and *smokers*, this approach removed over 50% of the prediction bias while improving accuracy 5-11%.

$\ell_1$ regularization proved least reliable at removing bias subject to not harming accuracy. In many cases, it was unable to remove much bias (*glioma, micromass, PC/Mac*). On synthetic data $\ell_1$ gave the best bias reduction. Though it did perform admirably on several real datasets (*arcene, prostate, smokers*), even removing up to 40% of the bias on the prostate dataset, it was consistently outperformed by either the parity or expert method. Additionally, on the *colon* dataset, it made bias significantly worse (150%) for gains in accuracy.

**Deep networks.**  The results show that deep networks tend to have a less significant feature asymmetry than data used for logistic models, which we would expect to render the feature parity approach less effective. The results confirm this, although on CIFAR10 parity had some effect on bias and a proportional positive effect on accuracy. Influence-directed experts, on the other hand, continued to perform well for the deep models. While this approach generally had a greater effect on accuracy than bias for the linear models, this trend reversed for deep networks, where the decrease in bias was consistently greater than the increase in accuracy. For example, the 7.7% bias in the original CelebA model was reduced by approximately 98% to 0.2%, effectively eliminating it from the model's predictions. The overall effect on accuracy remained modest (0.3% improvement).

These results on deep networks are somewhat surprising, considering that the techniques described in Section 4 were motivated by observations concerning simple linear classifiers. While the improvements in accuracy are not as significant as those seen on linear classifiers, they align with our expectations regarding bias reduction. This suggests that future work might improve on these results by adapting the approach described in this paper to better suit deep networks.

**Acknowledgments**  This work was developed with the support of the National Science Foundation under Grant No. CNS-1704845, and the Air Force Research Laboratory under agreement number FA8750-15-2-0277. The U.S. Government is authorized to reproduce and distribute reprints for Governmental purposes not withstanding any copyright notation thereon. The views, opinions, and/or findings expressed are those of the author(s) and should not be interpreted as representing the official views or policies of the Air Force Research Laboratory, the National Science Foundation, or the U.S. Government.

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

## A  PROOFS

**Proposition 1** *Let $\mathbf{x}$ be distributed according to Equation 3, $y$ be Bernoulli with parameter $p^*$, $D$ be the Mahalanobis distance between the class means $\boldsymbol{\mu}_0^*, \boldsymbol{\mu}_1^*$, and $\beta = -D^{-1}\log(p^*/(1-p^*))$. Then the bias amplification of the Bayes-optimal classifier $h^*$ is:*

$$B_{\mathcal{D}}(h^*) = 1 - p^* - (1 - p^*)\Phi\left(\beta + \frac{D}{2}\right) - p^*\Phi\left(\beta - \frac{D}{2}\right)$$

**Proof.** Note that the Bayes-optimal classifier can be expressed as a linear weighted sum (Murphy, 2012) in terms of parameters $\hat{\mathbf{w}}, \hat{b}$ as shown in Equation 8.

$$\Pr[Y = 1 | X = \mathbf{x}] = (1 + \exp -(\hat{\mathbf{w}}^T\mathbf{x} + \hat{b}))^{-1} \tag{8}$$

$$\hat{\mathbf{w}} = \hat{\boldsymbol{\Sigma}}^{-1}(\hat{\boldsymbol{\mu}}_1 - \hat{\boldsymbol{\mu}}_0)$$

$$\hat{b} = -\frac{1}{2}(\hat{\boldsymbol{\mu}}_1 - \hat{\boldsymbol{\mu}}_0)^T\hat{\boldsymbol{\Sigma}}^{-1}(\hat{\boldsymbol{\mu}}_1 + \hat{\boldsymbol{\mu}}_0) + \log\frac{\hat{p^*}}{1 - \hat{p^*}}$$

The random variable $\mathbf{w}^T X$ is a univariate Gaussian with variance $\mathbf{w}^T\boldsymbol{\Sigma}\mathbf{w}$ and mean $\mathbf{w}^T\boldsymbol{\mu}_y$ when $Y = y$. Then the quantity we are interested in is shown in Equation 9, where $\Phi$ is the CDF of the standard normal distribution.

$$\Pr\left[\mathbf{w}^T X > -b \,\middle|\, Y = y\right] = 1 - \Phi\left(\frac{-b - \mathbf{w}^T\boldsymbol{\mu}_y}{\sqrt{\mathbf{w}^T\boldsymbol{\Sigma}\mathbf{w}}}\right) \tag{9}$$

Notice that the quantity

$$\mathbf{w}^T(\boldsymbol{\mu}_1 - \boldsymbol{\mu}_0) = (\boldsymbol{\mu}_1 - \boldsymbol{\mu}_0)^T\boldsymbol{\Sigma}^{-1}(\boldsymbol{\mu}_1 - \boldsymbol{\mu}_0)$$

is the square of the Mahalanobis distance between the class means.

$$-b - \mathbf{w}^T\boldsymbol{\mu}_0 = \frac{1}{2}\mathbf{w}^T(\boldsymbol{\mu}_1 - \boldsymbol{\mu}_0) - \log\frac{p^*}{1 - p^*}$$

$$= \frac{D^2}{2} - \log\frac{p^*}{1 - p^*}$$

$$-b - \mathbf{w}^T\boldsymbol{\mu}_1 = -\frac{1}{2}\mathbf{w}^T(\boldsymbol{\mu}_1 - \boldsymbol{\mu}_0) - \log\frac{p^*}{1 - p^*}$$

$$= -\frac{D^2}{2} - \log\frac{p^*}{1 - p^*}$$

Similarly, we can rewrite the standard deviation of $\mathbf{w}^T X$ exactly as $D$. Rewriting the numerator in the $\Phi$ term of (9),

$$(\mathbf{w}^T\boldsymbol{\Sigma}\mathbf{w})^{\frac{1}{2}} = \left((\boldsymbol{\mu}_1 - \boldsymbol{\mu}_0)^T\boldsymbol{\Sigma}^{-1}\boldsymbol{\Sigma}\boldsymbol{\Sigma}^{-1}(\boldsymbol{\mu}_1 - \boldsymbol{\mu}_0)\right)^{\frac{1}{2}}$$

$$= \left((\boldsymbol{\mu}_1 - \boldsymbol{\mu}_0)^T\boldsymbol{\Sigma}^{-1}(\boldsymbol{\mu}_1 - \boldsymbol{\mu}_0)\right)^{\frac{1}{2}}$$

$$= D$$

Then we can write $\Pr\left[\mathbf{w}^T X > -b\right]$ as:

$$(1 - p^*)\left(1 - \Phi\left(\beta + \frac{D}{2}\right)\right) + p^*\left(1 - \Phi\left(\beta - \frac{D}{2}\right)\right)$$

$$= 1 - (1 - p^*)\Phi\left(\beta + \frac{D}{2}\right) - p^*\Phi\left(\beta - \frac{D}{2}\right)$$

$\square$

| Dataset | Features | Instances |
|---------|----------|-----------|
| CIFAR10 | 2,352 ($3 \times 28 \times 28$) | 12,000 |
| CelebA | 116,412 ($3 \times 218 \times 178$) | 202,599 |
| Arcene | 10,000 | 200 |
| Colon | 2,000 | 62 |
| Glioma | 22,283 | 85 |
| Micromass | 1,300 | 571 |
| PC/Mac | 1,943 | 3,289 |
| Prostate | 5,966 | 102 |
| Smokers | 19,993 | 187 |
| Synthetic | 1,002 | 100 |

Table 2: Number of features and instances for each dataset used in the evaluation in Section 5.

**Corollary 1** *When* $\mathbf{x}$ *is distributed according to Equation 3 and* $p^* = 1/2$, $B_\mathcal{D}(h^*) = 0$.

**Proof.** Note that because $p^* = 1/2$, the term $\beta = 0$ in Theorem 1. Using the main result of the theorem, we have:

$$
\begin{aligned}
\Pr\left[\mathbf{w}^T X > -b\right] &= 1 - \frac{1}{2}\left[\Phi\left(\frac{D}{2}\right) + \Phi\left(-\frac{D}{2}\right)\right] \\
&= 1 - \frac{1}{2}\left[\Phi\left(\frac{D}{2}\right) + \left(1 - \Phi\left(\frac{D}{2}\right)\right)\right] \\
&= \frac{1}{2}
\end{aligned}
$$

The third equality holds because $\Phi$ has rotational symmetry about $(0, 1/2)$, giving the identity $\Phi(-x) = 1 - \Phi(x)$.

□

