# OpenReview forum: "Feature-Wise Bias Amplification"
_ICLR.cc/2019/Conference_

### Official Review · AnonReviewer3 · 2018-10-28
**some insights on predictive bias and weak features**

**Rating:** 6
**Confidence:** 5

**Review:**

update: The authors' feedback has addressed some of my concerns. I update my rating to 6.
=================
original:
This paper provides some new insights into classification bias. On top of the well known unbalanced group size, it shows that a large number of weak but asymmetry weak features also leads to bias. This paper also provides a method to reduces bias and remain the prediction accuracy.

In general, the paper is well written, but some description can be clearer. Some notation seems inconsistent. For example, D in equation (1) denotes the joint distribution (x,y), but it also refers to the marginal distribution of x somewhere else.

In the high level, I am not totally convinced of how significant the result is.  In particular, the bias this paper defines is on the probability (softmax) scale, but logistic regression is on logit scale--   not even aimed at the unbiasedness in the original scale. So the result in section 2 seems to be expected.  Given the fact that unbiasedness is not invariant under transformation, I am wondering why it should be the main target in the first place.

In the bias reduction methods in equation 5 and 6, both the objective function and the constraint are empirical estimations. Will it be too noisy to adapt to the high dimensional setting? On the other hand, adding some sparsity regularization improves prediction seems well known in practice.

I would also encourage the authors to have extended work both theoretically and experimentally.  The asymmetry feature is only illustrated by a single logistic regression. Is it a problem of weak features, or indeed a problem of logistic regression? What will happen in a more general case beyond mean-field Gaussian?  I would imagine in this simple case the authors may even derive the closed form expression to verify their heuristics.

Based on the evaluations above, I would recommend a weak reject.

---

> ### Author Response · Authors · 2018-11-13
> **Thank you for your thoughtful feedback**
>
> While logistic regression is often on the logit scale, we tried to consistently use the probability scale in our analysis and experiments. If the paper contains any inconsistencies on this matter, we would appreciate knowing where they appeared so that we can address them. However, we would like to better understand the reviewer’s concern about unbiasedness failing to be invariant under transformation, and how we could have otherwise targeted our approach to better address the problem. With additional details, we hope to be able to address your concern.
>
> In (7) (formerly 6), we are minimizing the bias of the model over the choices of alpha and beta subject to not harming accuracy. It is true that when optimizing, the bias and accuracy of the model are necessarily obtained via an empirical estimation, so it is possible that the alpha and beta chosen wouldn’t generalize well to the test data. We treated these as normal hyperparameters in our experiments. The numbers reported in Table 1 report the bias and accuracy on the test data, while the optimization problem from (7) was solved on the training data, so we are reasonably confident that in practice the optimal alpha and beta generalize well, even in high-dimensional settings.
>
> Our aim was to identify the phenomenon of feature-wise bias on a class of problems that are sufficiently controlled so that we can make reasonable conclusions about the source of the bias. In the general case, beyond mean-field Gaussian, it may be harder to identify the source of the bias, as many sources may be interacting at once (e.g., feature-wise, class-imbalance, correlated features, etc.). We believe the results in Table 1 shed some light on the general case, namely, the bias is typically in the direction of the feature imbalance, even when this is at odds with the prior bias (as is the case in prostate). Furthermore, on some of the datasets (arcene in particular), balancing the number of features was quite effective at removing bias while improving accuracy, suggesting that a reasonable portion of the bias was caused by feature asymmetry.

---

> > ### Comment · AnonReviewer3 · 2018-12-01
> > **Thank you for clarifying**
> >
> > The rewritten sections are much clearer. The comparison between LR/SVM & L-BFGS/ SGD is really impressive. The comparison between LR/SVM without SGD makes it even more interesting to identify when bias asymmetry will be linked to bias, and so that when feature re-balancing helps.
> >
> > "namely, the bias is typically in the direction of the feature imbalance, even when this is at odds with the prior bias (as is the case in prostate)." I am confused. prostate has asymm<0.5 and bias>0.  Is it the same direction?

---

> > > ### Author Response · Authors · 2018-12-05
> > > **Thank you for following up**
> > >
> > > Apologies; you are correct that the direction of the bias is not immediately clear from Table 1, as Table 1 reports absolute bias (since this is what we would like to minimize). In our experiments we observed that the bias was in fact in the same direction as the feature asymmetry in prostate (i.e., bias with sign is -47.3); while we do not highlight this fact in the paper specifically, we will update the table to include the sign of the bias so the direction agreement is also clear.

---

### Official Review · AnonReviewer2 · 2018-11-02
**Well-motivated paper with a good balance of novel insight and practical methods**

**Rating:** 7
**Confidence:** 4

**Review:**

Summary:

In this paper the authors identify a specific source of marginal class probability bias that occurs when using logistic regression models. Using synthetic and real datasets they demonstrate this bias and explore characteristics of the data that exacerbate the issue. Finally, they propose two methods for correcting this bias in logistic regression models and neural network models with logistic output layers and evaluate these methods on several benchmark datasets.

Review:

Overall, I found the paper well-written, the problem well-motivated, and the proposed methods clear and reasonable. While I have a few concerns about presentation and experimentation, these are issues that can easily be remedied and I recommend acceptance.

Major comments:

- The authors repeatedly say that gradient descent is the cause of the bias amplification (e.g. Section 2.2 title, "...features that are systematically overestimated by gradient descent.", "... i.e., a consequence of gradient descent's inductive bias.", "... gradient descent may lead to systematic bias..."). The inductive bias they describe is coming from the use of logistic regression, not the use of gradient descent. Specifically, a logistic regression model has a convex likelihood, which means that regardless of what algorithm is used to maximize the likelihood, it should converge to the same point. In fact, most off-the-shelf implementations of logistic regression do not use vanilla gradient descent. Further, gradient descent may be used to estimate the parameters of any number of models which may or may not have the same inductive bias the authors describe.

- I thought the related work section was well-written and would strongly recommend moving it to the beginning of the paper as it motivates the entire problem. I also think it could be helpful to ground the technical definitions of bias amplification in a meaningful example.

- I think that the experimental setup for comparing \ell_1 regularization to the proposed feature selection methods is not quite fair. In particular, the hyperparameters of the "expert" method are selected to minimize bias subject to the constraint that loss not increase. In contrast, the \ell_1 regularization hyperparameter is selected purely to minimize bias. Instead, I would select the \ell_1 regularization hyperparameter in the same way as the expert method, that is, to minimize bias subject to a constraint on loss. In general, I think hyperparameters should be selected using the same criterion for all methods.

- The authors make a point of highlighting results on the "prostate" which showed a large increase in accuracy along with a large decrease in bias. I think the paper would benefit from some exploration of why this happened. Specifically, it would be valuable to answer the question: what are the properties of the "prostate" dataset that make this method so effective and are these properties general and identifiable a priori?

- Section 2, paragraph 2, line 5: The stated goal in this paragraph is "minimizing 0-1 loss on unknown future i.i.d. samples". As stated in the introduction, this is, in fact, not the goal. The goal is to minimize loss while also minimizing bias. A larger criticism that I would have of this work is: if minimizing bias is a first order goal, then why are we using empirical risk minimization in the first place? Put another way, why use post-hoc correction for an objective function that does not match our actual stated goals rather than using an objective function that does?

Minor comments:

- Section 1, paragraph 4, line 2: "Weak" is not clearly defined here. Is it different than "moderately-predictive"?

- Section 2.1, last paragraph, line 1: I understand what the authors are saying when they say "Bias amplification is unavoidable", but it is avoidable by changing our objective function. I would consider rewording this statement to something like "Using an ERM objective will lead to bias amplification when the learning rule..."

- Equation 4: I believe h should be changed to f in this equation.

- Equation 6: L is not defined anywhere.

- Table 1: As defined in equation 1, B_D(h_s) should be between 0 and 1. Also, the accuracy results for the glioma dataset have the wrong result in bold.

- Section 4, methodology paragraph, line 5: forthe --> for the

- Section 5, paragraph 5, lines 5-6: Feature selection is not used "only to improve accuracy". For example, Kim, Shah, and Doshi-Valez (2015) use feature selection to improve interpretability (https://beenkim.github.io/papers/BKim2015NIPS.pdf).

---

> ### Author Response · Authors · 2018-11-13
> **Thank you for your thoughtful feedback**
>
> We agree that the results we have presented do not indicate that SGD is the exclusive cause of the bias-inducing behavior examined in the paper. We note that LR will, given enough data, converge to the Bayes-optimal classifier, and because the data used in Figure 2 has an unbiased prior, we would expect no bias in the predictions according to Thm. 1. However, we posit that feature-wise bias occurs when the learner has not seen enough data to converge. While we observed this consistently with models trained using SGD, it may indeed happen when other methods are used to learn the coefficients from insufficient data. On the other hand, different methods may yield different models when training ends prior to convergence.
>
> We have updated the paper with additional results that shed more light on the sources of bias in linear models. Figure 3 in the appendix depicts the bias of classifiers trained using the same data as in Figure 2, including LR trained with either L-BFGS or SGD, linear SVM trained with either SMO or SGD, and SGD using modified Huber and squared hinge losses. In short, while LR trained with L-BFGS does exhibit some bias, it is not as pronounced or consistent as it is in models trained with SGD, whereas all the models trained with SGD exhibited nearly identical bias trends. In slightly more detail, LR trained without SGD was less sensitive to the number of weak features, i.e., there was less bias than LR trained with SGD until there was a sufficiently high number of weak features, and even then, the effect was not as strong. Furthermore, SVM trained without SGD exhibited essentially no such bias, while SVM trained with SGD exhibited the same bias as LR with SGD. These results suggest that while the bias-inducing behavior may occur when other methods are used, they consistently follow from the use of SGD.
>
> Thank you for your feedback on the related work section, we have moved it to the front of the paper as suggested.
>
> Thank you for your comment about L1 versus experts method parameters--upon review, the wording in the experiments section is not clear. We did use the same procedure for finding the hyperparameter for L1 regularization as for the experts technique, i.e. we optimized for minimizing bias subject to the constraint that accuracy should not decrease from the original model. You may have noticed that on the glioma dataset, the accuracy goes down for L1. We conjecture that this is caused by the hyperparameter not generalizing well to the test data, as we evaluated hyperparameters on the training data. We have updated the writing in Section 4 to clarify this.
>
> It’s not immediately clear what distinguishes the prostate data from the others, but upon inspection, prostate has a rather high Mahalanobis distance between classes compared to many of the other datasets. This might suggest there was a lot of room for improvement on this dataset (i.e., the bias was largely preventable because the classes are well-separated). Like most of the other datasets, prostate had a huge disparity in the number of data points (small) to features (large), so it is perhaps unsurprising that despite having the classes fairly well-separated in its feature space, a model with no regularization was unable to generalize well on it. Furthermore, prostate was the only dataset for which the feature disparity opposed the prior bias (and moreover the bias went in the direction of the features rather than the prior), so perhaps the feature-wise bias was the most significant source of bias in this example. It may be an interesting avenue for future work to investigate whether, e.g., Mahalanobis distance between classes, is a good predictor for the effectiveness of our techniques on real data.
>
> In Section 3 (previously Section 2), paragraph 2, we state the goal (minimizing 0-1 loss) of the “standard binary classification problem,” not the overall goal of our paper. In fact, our goal is not exactly to generally minimize bias along with loss; we note that there are multiple possible sources of bias, only some of which are avoidable when optimizing accuracy. Namely, as stated in Theorem 1, an optimal classifier may necessarily be biased in some cases. Our goal is to remove bias that is not “necessary” in this way, which is not easily captured by additional terms in the training objective. Our work identifies feature-wise bias as one type of preventable or “unnecessary” bias, and attempts to remove it in a targeted fashion with post-hoc feature selection. In other words, we want our model to be no more biased than the most accurate predictor, which may still have some bias according to Theorem 1 (but we consider this bias unavoidable because it can be considered equally problematic to sabotage accuracy in order to reduce bias).
>
> Thank you for your minor comments as well, we have addressed them in the updated paper.

---

> > ### Comment · AnonReviewer2 · 2018-11-21
> > **Thank you for clarifying**
> >
> > RE: The source of bias - In light of this comment, I think you need to be *very* careful about how you describe the sources of bias in the paper. For example, the second paragraph of section 3.2 in the updated paper says "Logistic regression models make fewer assumptions about the data and are therefore more widely-applicable, but as we demonstrate in this section, this flexibility comes at the expense of an inductive bias that can lead to systematic bias in predictions." I read this as implying that LR is the source of the bias which your experiments seems to suggest it isn't. As another example, the last paragraph of section 3.2.1 in the updated paper says "Figure 2c suggests that overestimation of weak features is precisely the form of inductive bias exhibited by gradient descent when learning logistic classifiers." Your analysis suggests that it is largely due to SGD rather than general gradient descent so I would replace any mention of "gradient descent" with "SGD". In light of the updates, I think this paper would be a lot stronger if it focused on identifying and describing the source of the bias (this would be a much more general result), but is still worth publishing if the authors are careful about the scope of their claims.
> >
> > RE: "it can be considered equally problematic to sabotage accuracy in order to reduce bias" - I would argue that this is exactly what we want to do in many settings where we care about bias. For example, we should be willing to sacrifice accuracy in recidivism prediction in order to avoid racial bias. A focus on accuracy first is exactly the mindset that has led to algorithmic fairness becoming a serious issue.

---

> > > ### Comment · AnonReviewer2 · 2018-11-21
> > > **Clarification**
> > >
> > > I think I should clarify a bit what I mean when I say "this would be a much more general result" and why I think it would make the paper better. As I see it, the main contribution of the paper is an observation that weak features lead to bias amplification in logistic regression models when the parameters are estimated using SGD. To be clear, I think this is a valuable observation in and of itself, and the authors are rigorous in confirming and describing this observation (section 3.2 of the updated paper); however, the scope of this observation is unclear. For example, does bias amplification occur in any setting with weak features regardless of the model and optimization method used (I assume not, but this is not evaluated)? Does bias amplification occur in any classification model trained using SGD or only linear models? At the core of these questions is the "why" question: "what are the properties of LR, SGD, or their combination that lead to bias amplification in the presence of weak features?" Answering this question would be a more general result because it would let us identify the problem in other settings without the need for experimentation and would allow us to propose fixes that are based on addressing the root cause rather than heuristics.

---

> > > > ### Author Response · Authors · 2018-11-27
> > > > **Thank you for following up**
> > > >
> > > > Thank you for your further feedback on the story of the paper. To answer your specific questions: we do not believe that the form of bias amplification identified in this paper as “avoidable” occurs whenever unbalanced features are present, as we observed that linear SVM models trained using SMO do not exhibit it (Figure 3 in the appendix); this form of bias amplification does not just occur in linear models, as we observed it in the two deep convolutional networks presented in our evaluation (Table 1). We agree that a more general result that pinpoints why SGD overestimates weak features is interesting and an avenue of future work. We see the contributions in this paper as a necessary first step towards answering these more general “why” questions, and look forward to further analysis of this phenomenon as future work.
> > > >
> > > > We appreciate your suggestions on framing our claims, and will revise the writing accordingly prior to future submission or publication to ensure that our precise claims are clear and not overstated.
> > > >
> > > > We certainly agree that in some cases we may reasonably want to sacrifice accuracy for bias. In these cases we might, e.g., use a notion of fairness to guide how we handle the trade-off. It was not our intention to take a specific position on this trade-off, or to weigh in on defining fairness. Rather, we aim to point out that in the case of “avoidable” bias, there is no such trade-off, as bias and accuracy are not in conflict. Mitigating feature-wise bias may be used in conjunction with other techniques in the context of fairness.

---

> > > > > ### Comment · AnonReviewer2 · 2018-11-30
> > > > > **Very good clarification**
> > > > >
> > > > > I really like the statement: "Rather, we aim to point out that in the case of “avoidable” bias, there is no such trade-off, as bias and accuracy are not in conflict." It's entirely possible that I just missed it, but I think a statement of this type and some discussion of the broader trade-offs would go very well in the introduction. People are thinking a lot about this issue and I think this paper makes a good argument that, in fact, there may be some low hanging fruit where there is basically no trade-off at all.

---

### Official Review · AnonReviewer1 · 2018-11-03
**Interesting result, need more comparison in the experiment section, need more explaining of related work**

**Rating:** 6
**Confidence:** 4

**Review:**

In this paper, the authors studied bias amplification. They showed in some situations bias is unavoidable; however, there exist some situations in which bias is a consequence of weak features (features with low influence to the classifier and high variance). Therefore, they used some feature selection methods to remove weak features; by removing weak features, they reduced the bias substantially while maintaining accuracy (In many cases they even improved accuracy).  Showing that weak features cause bias is very interesting, especially in their real-world dataset in which they improved bias and accuracy simultaneously.


My main concerns about this paper are its related work and its writing.
Authors did a great job in reviewing related work for bias amplification in NLP or vision.
However, they studied bias amplification in binary classification, in particular, they looked at GNB; and they did not review the related work about bias in GNB.  I think it is clear that using MAP causes bias amplification. Therefore, I think changing theorem 1 to a proposition and shifting the focus of the paper to section 2.2 would be better. Right now, I found feature orientation and feature asymmetry section confusing and hard to understand. In the paper, the authors claimed bias is a consequence of gradient descent’s inductive bias, but they did not expound on the reasoning behind this claim. Although the authors ran their model on many datasets, there is no comparison with previous work. So it is hard to understand the significance of their work. It is also not clear why they don’t compare their model with \ell_1 regularization in CIFAR.


Minor:

Paper has some typos that can be resolved.
Citations have some errors, for example, Some of the references in the text does not have the year, One paper has been cited twice in two different ways, For more than two authors you should use et al., sometimes \citet and \citep are used instead of each other.
Authors sometimes refer to the real-world experiment without first explaining the data which I found confusing.

---

> ### Author Response · Authors · 2018-11-13
> **Thank you for your thoughtful feedback**
>
> Thank you for your comments regarding the previous work section. We have included a more in-depth comparison to other work around bias in GNB in our update to the paper.
>
> We have updated Section 2.2 (now Section 3.2) with a more precise description of the data used in that section, which was constructed to exemplify the feature asymmetry we describe. We hope that it clears up some of the confusion in that part of the paper, and are willing to revise with additional clarifications if needed.
>
> Regarding the claim that bias follows from an inductive bias of SGD, the argument is that because we see bias when we train SGD-LR in a setting where the Bayes-optimal classifier would have no bias, the bias cannot be explained by Theorem 1 (i.e., as bias that is inevitable when optimizing accuracy), hence we conclude the bias must have been caused by the learning rule (SGD-LR). While the inductive bias may not be uniquely attributable to SGD, and instead may be a consequence of using LR regardless of how the coefficients were obtained, we found that LR models trained on the same data using other methods, such as L-BFGS, did not result as much consistent bias as LR trained with SGD. Moreover, training with SGD using other loss functions, such as hinge, modified-Huber, and perceptron, resulted in the same bias characteristics as shown in Figure 2. Thus, linear classifiers trained with SGD consistently show the inductive bias we describe, whereas comparable classifiers trained using other methods may not. We have included an additional figure (Fig. 3 in the appendix) that details these results.
>
> In our experiments we compare our feature selection method targeted at feature-wise bias to L1 regularization. We are not aware of other feature selection methods intended to mitigate the bias we target in the paper, but are willing to include additional comparisons if there are comparable approaches that we missed.
>
> We additionally added results for L1 regularization on CIFAR. In general, L1 is harder to apply to the deep network scenarios because training takes a long time, making the hyperparameters hard to tune.
>
> Thank you also for your formatting comments; we have addressed them in the updated version of the paper.

---

> > ### Comment · AnonReviewer1 · 2018-12-03
> > **Thanks for clarifying**
> >
> > Thanks for your answer and the revision. The writing and the structure of the paper are much better now. I still have two issues with the paper.
> >
> > 1) You’ve suggested asymmetry of the features is one of the reasons that SGD leads to systematic bias (e.g., you have written: “When the data is distributed asymmetrically with respect to features’ orientation towards a class, gradient descent may lead to systematic bias”). I’m wondering what the reason behind this claim is?
> >
> > In your synthetic dataset (Figure 2) all the features are asymmetric; and, you did not study presence of bias when there are lots of symmetric weak features (e.g., instead of \miu_1 = (1,0,1,1,...1); assuming \miu_1 = (1,0,1,-1,1,-1,…, 1)
> >
> > In the experiment section building upon this claim, you introduced feature parity to mitigate bias; however, feature parity does not have a very good performance in comparison to the other method (Experts). So, I’m not sure how much I can believe that asymmetry causes bias.
> >
> >
> > 2) I took a look at the statistics of some of the datasets in your experiment (datasets from Li et al., 2016), and I realized in some datasets there are 10X to 100X more features than instances. E.g., the prostate has 100 instances while 50K features (I would suggest having a small table about statistics of the datasets).
> > Given these statistics, it is somehow clear that overfitting happens in the training; therefore, improvement of the accuracy is not surprising (note that in the prostate increment in accuracy causes the reduction in bias; all the error (all the 10%) are still toward one of the classes).
> >
> > I’m wondering if there is anything special about your feature selection methods. I mean if I use other feature selection methods how do they perform regarding the bias reduction? As I checked (in Li et al., 2016), some other feature selection methods increase the accuracy comparable or sometimes better than your methods.
> >
> > Again, I would like to mention that I really liked the idea of showing weak features cause systematic bias; and I liked that you experimentally showed even with p*=0.5, SGD leads to systematic bias.
> >
> >
> > Minor:
> > Is the below equation right?
> > Bias <= 100 – accuracy
> > why does this not hold for prostate dataset?

---

> > > ### Author Response · Authors · 2018-12-05
> > > **Thank you for following up**
> > >
> > > Thank you for your continued feedback. We ran the experiment suggested, where \mu_1 = (1,0,1,0,1,...,0), and this results in no systematic bias (with a setup similar to that of Figure 2(a), but with 200 weak features - 100 per class - and N=1000, the average bias over 100 trials was 0.00031, which would round to 0.0% using the same precision as in Table 1). We believe this result makes sense: since permuting the order of the features does not affect the result, it would not be possible to have bias when the features are entirely symmetric, because the orientation of the features can be reversed simply by permuting them when there are the same number of features oriented in each direction. We show empirically that the weaker features are more likely to be overestimated by SGD, but without any asymmetry, we would expect that this would affect both classes equally in expectation.
> > >
> > > We agree that the asymmetry need not be precisely in the *number* of weak features, as it was in the synthetic data. For example, some weak features may be weaker than others, and there may be a disparity in the total strength of the features for each class. Thus, more complicated cases may be slightly harder to analyze. In this vein, on the real data, feature parity is likely often overly simple, as it doesn’t necessarily balance the total strength of the features for each class. Experts are more targeted towards balancing the strength of the features rather than only the number, which was likely more appropriate for most of the real datasets.
> > >
> > > Thank you for your suggestion regarding the additional information on the datasets. In Section 5 we briefly note that we selected datasets based on high feature dimensionality, but we can also include a table with further details in the appendix.
> > >
> > > We agree that overfitting likely happens to some extent on these datasets during training. What is interesting is that our techniques are post-hoc, meaning that models are not retrained following feature selection, they are simply pruned. Intuitively, this could perhaps be interpreted to mean the strong features were learned well and the overfitting happens primarily with the weak features. Aside from being interesting from the perspective of understanding the bias/overfitting, our techniques are specifically targeted towards removing bias when improving accuracy, while we see, e.g., L1 cannot typically accomplish both of these goals. Comparison to some of the methods in Li et al. would be interesting in the context of bias. However, even if the performance of other methods were the same, our methods could still be preferable in some contexts because they are easily and quickly applied post-hoc, and are easily extended to deep networks.
> > >
> > > We apologize, the equation bias + accuracy <= 100 is correct; upon reviewing our data it appears we rounded 0.0980 incorrectly to 0.0100 when writing it into Table 1 (the bias and accuracy were 0.0980 -> 9.8 and 0.9019 -> 90.2 respectively). We will update Table 1 with this correction.

---

### Meta-Review · Area_Chair1 · 2018-12-15
**well written paper with theoretical and experimental validation**

**Confidence:** 3
**Recommendation:** Accept (Poster)

**Metareview:**

The authors identify a source of bias that occurs when a model overestimates the importance of weak features in the regime where sufficient training data is not available. The bias is characterized theoretically, and demonstrated on synthetic and real datasets. The authors then present two algorithms to mitigate this bias, and demonstrate that they are effective in experimental evaluations.
As noted by the reviewers, the work is well-motivated and clearly presented. Given the generally positive reviews, the AC recommends that the work be accepted. The authors should consider adding additional text describing the details concerning Figure 3 in the appendix.